# Doctor Referral of Overweight People to a Low-Energy Treatment (DROPLET) in primary care using total diet replacement products: a protocol for a randomised controlled trial

Susan A Jebb, Nerys M Astbury, Sarah Tearne, Alecia Nickless, Paul Aveyard

► Prepublication history and additional material are available. To view these files please visit the journal online (http://dx.doi.org/10.1136/bmjopen-2017-016709).

Nuffield Department of Primary Care Health Sciences, University of Oxford, Oxford UK

**Correspondence to**
Dr Nerys M Astbury;
nerys.astbury@phc.ox.ac.uk

## ABSTRACT

**Introduction** The global prevalence of obesity has risen significantly in recent decades. There is a pressing need to identify effective interventions to treat established obesity that can be delivered at scale. The aim of the Doctor Referral of Overweight People to a Low-Energy Treatment (DROPLET) study is to determine the clinical effectiveness, feasibility and acceptability of referral to a low-energy total diet replacement programme compared with usual weight management interventions in primary care.

**Methods and analysis** The DROPLET trial is a randomised controlled trial comparing a low-energy total diet replacement programme with usual weight management interventions delivered in primary care. Eligible patients will be recruited through primary care registers and randomised to receive a behavioural support programme delivered by their practice nurse or a referral to a commercial provider offering an initial 810 kcal/d low-energy total diet replacement programme for 8 weeks, followed by gradual food reintroduction, along with weekly behavioural support for 24 weeks. The primary outcome is weight change at 12 months. The secondary outcomes are weight change at 3 and 6 months, the proportion of participants achieving 5% and 10% weight loss at 12 months, and change in fat mass, haemoglobin A1c, low-density lipoprotein cholesterol and systolic and diastolic blood pressure at 12 months. Data will be analysed on the basis of intention to treat. Qualitative interviews on a subsample of patients and healthcare providers will assess their experiences of the weight loss programmes and identify factors affecting acceptability and adherence.

**Ethics and dissemination** This study has been reviewed and approved by the National Health ServiceHealth Research Authority (HRA)Research Ethics Committee (Ref: SC/15/0337). The trial findings will be disseminated to academic and health professionals through presentations at meetings and peer-reviewed journals and to the public through the media. If the intervention is effective, the results will be communicated to policymakers and commissioners of weight management services.

**Trial registration number** ISRCTN75092026.

## Strengths and limitations of this study

► This study is the largest randomised controlled trial to date of a low-energy total diet replacement programme for weight management in routine primary care.
► This intervention is based on a model of care where general practitioners refer patients to a programme delivered in the community by a commercial provider using non-National Health Service staff, which, if successful, could be readily adopted into practice without the need for specialist training for the primary care workforce.
► The primary outcome is weight at 1 year. Although this is 9 months after the low-energy total diet replacement, epidemiological evidence suggests that any weight lost will continue to be regained beyond 1 year.
► The intention of obesity treatment programmes is to improve long-term health, but this study does not include morbidity or mortality outcomes.
► Longer term follow-up data would be helpful to better estimate the longer health impact and cost-effectiveness of the intervention.

## INTRODUCTION

The prevalence of obesity worldwide has more than doubled since 1980.[1] According to the latest estimates from the WHO, more than 1.9 billion adults were overweight, of whom 600 million were obese, representing 39% and 13% of the world's adult population, respectively.[2] Obesity is associated with premature mortality,[3] but also substantial morbidity, including significantly increased risks of diabetes, cardiovascular disease and most non-smoking-related cancers, as well as physical impairments linked to excess weight such as breathlessness, joint problems and back pain.[4] Collectively this creates a burden of ill-health and reduced quality of life for individuals, additional treatment costs to

the National Health Service (NHS) and reductions in economic productivity.[5] While high priority must be given to prevent future cases of obesity, in the short term, there is a pressing need to identify effective interventions to treat established obesity. Research has shown that even modest reductions in weight can bring significantly reduced risks of disease. For example, in the US Diabetes Prevention Program (DPP), individuals randomised to an intensive lifestyle intervention lost 7 kg by the end of the first year. Although some of this weight was regained, the intensive lifestyle group remained 4 kg lighter than the usual care group at 4 years, and this reduced the incidence of diabetes by 58% relative to usual care,[6] with benefits persisting to at least 15-year follow-up despite weight regain.[7]

Primary care is an important setting for weight management interventions to reduce multimorbidity. However, although a number of interventions have been shown to be effective in intensive research studies, this success has not always been replicated in routine settings. For example, there was no significant reduction in weight when a weight loss programme adapted from the DPP was delivered by primary care teams.[8] Our recent review of interventions suitable for use in routine care[9] and a second review, using slightly different inclusion criteria, of interventions specifically delivered in primary care[10] both concluded that behavioural weight management interventions led by primary care practitioners were ineffective. This may relate in part to the complexity of advice needed for successful dietary change and the need for frequent contact to provide support, which exceeds the capacity of routine primary care systems. However, although a number of interventions have been shown to be effective in intensive research studies, this success has not always been replicated in routine settings. General practitioner (GP) referral to a commercial provider offering group-based support is an effective option for weight management in primary care, and our meta-analysis showed a mean reduction in weight of 2.3 kg over no intervention at 1 year.[9] However, greater weight losses would be expected to bring greater health gains.

Very low-energy diets (VLEDs) have been used for weight loss over many years in specialist settings. A VLED is defined as a diet providing ≤800 kcal a day, based on the use of specially formulated products designed as the sole source of nutrition during periods of total diet replacement. When used as directed, these formula products meet 100% of the dietary reference values for vitamins, minerals and trace elements for healthy, weight-stable people and are enriched with high biological-value protein. Although most contain some dietary fibre, a fibre supplement may also be recommended. A recent systematic review and meta-analysis of the available randomised controlled trials showed that behavioural weight management interventions incorporating a VLED led to 3.9 kg greater weight loss at 1 year compared with intensive specialist-delivered behavioural programmes.[11] However, most of the trials included in this review were small, typically including only 50–100 participants who were treated by obesity specialists, and many trials had methodological limitations.

UK guidance from the National Institute for Health and Care Excellence (NICE) recommends that VLEDs may only be used for a maximum of 12 weeks in people who have a clinical need to lose weight rapidly, such as prior to a knee replacement surgery or those seeking fertility services, but recommends against their routine use to manage obesity.[12] Clinical guidance in the USA does not recommend the routine use of VLEDs, but rather suggests that their use '*may* be reasonable in limited circumstances, but only when provided by trained practitioners in a medical care setting where medical monitoring and high intensity lifestyle intervention can be provided'.[13]

Nevertheless, there has been growing interest in the potential for routine use of weight loss programmes similar to traditional VLEDs, in so far as they incorporate a period of total diet replacement using specially formulated products as the sole source of nutrition, but where the energy content is more than 800 kcal/day but less than 1200 kcal/day. The NICE guidelines suggest that this type of low-energy diet could be considered for weight management, providing care is taken to ensure they are nutritionally complete.[12] There is one observational report (n=91) on the use of these low-energy total diet replacement programmes in primary care which found that 64% of participants completed the 810 kcal/day dietary programme, defined as either 12 weeks or reaching 20 kg weight loss, with a mean weight loss of 16.9 kg (SD=6.0 kg). One-third of participants starting the programme maintained a weight loss of ≥15 kg at 12 months.[14] A large randomised controlled trial, the DiRECT (Diabetes Remission Clinical Trial) study, is currently underway to investigate whether this type of low-energy total diet replacement programme can be used to treat type 2 diabetes among people who are also overweight.[15] It will compare the health effects of the current best-available type 2 diabetes care with those achieved through weight management based on a low-energy total diet replacement programme. While this will provide important mechanistic evidence on the links between weight loss and diabetes risk, it will be delivered by NHS staff, whereas the present study will test the effectiveness of referral outside the NHS to a commercial provider.

To fill this evidence gap, we will conduct a randomised controlled trial to specifically test the effectiveness of a GP referral to a community-based low-energy total diet replacement programme for patients who are obese and likely to benefit from weight loss. It will assess the clinical effectiveness of a weight loss intervention by measuring weight loss and the change in biomarkers of cardiovascular risk at 12 months relative to weight loss advice provided by practice nurses. This comparator is intended to represent 'usual care', although in practice most patients who are obese are not offered support to lose weight.

The context for this trial follows the established model for GP referral to community group-based weight loss programmes.[16] This uses the generic authority and credibility of health professionals to motivate patients to consider weight management and the specialist knowledge of the commercial provider to guide the intervention and offer frequent contact and behavioural support to the patient. If successful, it will provide another option for weight management that can be offered to patients in primary care, and GPs will be able to guide patients towards the treatment that best fits their circumstances and preferences. This trial will specifically test whether a partnership between GPs and providers will allow for the safe provision of low-energy total diet replacement programmes even for patients with multimorbidity who may gain the greatest benefits from such interventions but who may also need clinical oversight and adjustments to some of their medications as they lose weight. It will provide the opportunity for qualitative research to investigate the perspectives of patients and healthcare practitioners on this type of treatment.

### Objective

The aim of the Doctor Referral of Overweight People to a Low-Energy Treatment (DROPLET) trial is to determine the clinical effectiveness, feasibility and acceptability of referral to a low-energy total diet replacement programme compared with usual weight management interventions in primary care.

## METHODS
### Design and setting

The study will take place in general practices in England. The study is designed as an individually randomised, two-arm and parallel group superiority trial with the primary endpoint as objectively measured changes in body weight from baseline to 12 months. Due to the nature of the intervention, it will not be possible to blind participants, clinicians or some of the study team to the treatment allocation after randomisation.

### Recruitment

Around 10 general practices will be identified to take part through the clinical research networks. Recruited practices will be asked to conduct a search of their electronic health records in order to identify suitable patients for the DROPLET study. As a result of this search, eligible patients will be sent an invitation letter from their GP as part of a staggered mailout. Patients will be encouraged to call the research team if they are interested in taking part.

GPs may also identify eligible patients during routine consultations. The GP will provide the patient with an invitation letter and suggest that the patient ring the study team. The study team will provide the potential participants with information on what taking part in the study will entail, and an initial assessment of suitability to take part. Those who make contact and self-report meeting

the eligibility criteria will be scheduled for a baseline/enrolment appointment.

### Inclusion criteria

► participant is willing and able to give informed consent for participation in the study
► aged 18 years or above
► body mass index (BMI) $\geq 30 \, kg/m^2$
► likely to benefit from weight loss in the GP's opinion.

### Exclusion criteria

► unable to understand English
► currently or recently (within 3 months of study entry) attended a weight management programme or currently participating in another weight loss study
► had bariatric surgery or scheduled bariatric surgery
► pregnant, breast feeding or planning to become pregnant during the course of the study
► receiving insulin therapy
► heart attack or stroke within the last 3 months
► heart failure of grade II New York Heart Association and more severe
► angina, arrhythmia, including atrial fibrillation or prolonged QT syndrome
► taking monoamine- oxidase inhibitor (MAOI) medication
► taking anticoagulant medication (eg, warfarin)
► taking varenicline (smoking cessation medication)
► chronic renal failure of stage 4 or 5
► active liver disease (except non-alcoholic fatty liver disease (NAFLD), a history of hepatoma or within 6 months of onset of acute hepatitis
► people having active treatment for cancer other than skin cancer treated with curative intent by local treatment only, or people taking hormonal or other long-term secondary prevention treatment after initial cancer treatment
► active treatment or investigation for possible or confirmed gastric or duodenal ulcer; maintenance treatment with acid suppression is not a contraindication
► porphyria
► scheduled for surgery within 12 months
► a member of household is already enrolled in the study
► unwilling to provide blood samples
► patients that the GP judges not able to meet the demands of either treatment programme or measurement schedule; this may include severe medical problems not listed above or severe psychiatric problems including substance misuse that make following the treatment programme or adhering to the protocol unlikely.

### Participant flow

The baseline/eligibility assessment will be scheduled with a practice nurse or healthcare assistant at their own GP practice, where informed consent for participation in the study will be obtained before eligibility will be formally

assessed. After demographic information and all baseline measurements have been collected, the participant will be randomised to the allocated treatment group using the online randomisation system. The patients' own GP will be notified by letter of the enrolment and randomisation of their patient, so that it may be documented on their medical record. Participants allocated to the low-energy total diet replacement programme and taking medications for type 2 diabetes, hypertension or high cholesterol will have their medications reviewed by a prescribing member of the clinical care team, usually the GP or trained nurse prescriber. During this medication review, the clinician will decide what changes to medications are required at the time the participant commences the low-energy total diet replacement programme, with guidance provided by the study team (see online supplementary figure 1). In addition, participants randomised to the low-energy total diet replacement group and who take antihypertensive medications will be provided with a home blood pressure monitor and asked to record blood pressure once daily during the weight loss phase (weeks 1–12). These readings can be used to guide clinicians with any further changes in hypertension medications.

All participants will be invited to attend a 4-week follow-up appointment with the practice nurse. The main purpose of the visit is a clinical review of medication, including any adjustments required. Any changes in medication will be recorded on the concomitant medication log. Participants will be invited to attend further follow-up visits with a member of the trial team at the GP practice at 12 weeks, 6 months and 12 month following randomisation. Participant flow through the study is outlined in figure 1.

### Sample size
The total number of participants to be recruited for this study is 270. This is based on a sample size calculation for the primary outcome using equal variance independent samples t-test assuming a difference between groups at 12 months of 4 kg with an SD in both groups of 9 kg, obtained from a meta-analysis of published studies.[11] The sample size has been inflated by 20% to account for attrition, and assumes 90% power and two-sided alpha of 5%.

### Randomisation
All eligible, consenting participants will be randomised with an allocation ratio of 1:1 to low-energy total diet replacement or usual care programmes using an online programme, which reveals group allocation as per a computer-generated randomisation list. The randomisation criteria will be validated by an independent statistician. Allocation will be stratified by GP practice and baseline BMI ($\leq$35 kg/m$^2$ or >35 kg/m$^2$) using stratified block randomisation with randomly varying block sizes of 2, 4 and 6. The randomisation software ensures full allocation concealment, with the allocation group only revealed to the person performing the randomisation once a study identifier and required stratification details have been entered.

## Interventions
### Low-energy total diet replacement
The programme offered to participants randomised to the active intervention will be provided by Cambridge Weight Plan, Northants, UK.

Following randomisation participants allocated to this group will be referred to a local Cambridge Weight Plan counsellor who will invite the participant to attend regular appointments for 24 weeks. These appointments consist of motivational support, encouragement, reassurance and problem-solving. All counsellors attend a 1-day in-person training course covering screening for suitability, nutrition, behavioural approaches and medical monitoring. They must pass an accreditation examination before they are allowed to deliver the programme in the community. Thereafter, they have a yearly training updates, a nominated sponsor (experienced counsellor) and access to an online chat forum for sharing queries. Cambridge Weight Plan has a healthcare professional available for the counsellors to consult for advice on specific medical and nutritional queries. Counsellors delivering the intervention for the purposes of this trial received short trial specific training before being allocated study participants.

During the first 12 weeks the participant will meet with their counsellor weekly. Patients will be asked to follow a programme based on using formula meal replacement products (soups, shakes and bars) and milk comprising 810 kcal/day (3389 kJ/day). For the first 8 weeks, patients will be advised to replace *all* their usual foods and drinks with four of the formula products daily: 750 mL of skimmed milk, 2.25 L of water or other non-calorific drinks and a fibre supplement (total diet replacement stage). During the first 2 weeks, the formula products will be limited to liquid products (soups and shakes), but from week 3 onwards participants will have the option to include meal replacement bars as part of the formula product allowance. After 8 weeks there will be a 4-week stepwise reduction in the use of formula meal replacement products and a gradual reintroduction of food-based meals. The weight maintenance phase from week 12 to 24 participants attend monthly appointments at 16, 20 and 24 weeks, during this phase participants are advised to consume only one formula product a day, with the remainder of the diet to consist of self-selected food. This weight maintenance phase will include a recommendation to return to the total diet replacement stage for periods of up to 4 weeks if participants regain 1 kg or more than their weight measured at 12 weeks.

All consultations with counsellors and formula products will be provided to participants by their nominated counsellor and will be free of charge for the first 24 weeks, after which the intervention will end. Participants in both groups will be free to choose whether or not to continue with the programme, but at their own cost.

### Comparator
The comparator intervention will consist of the usual weight management programme provided by a member

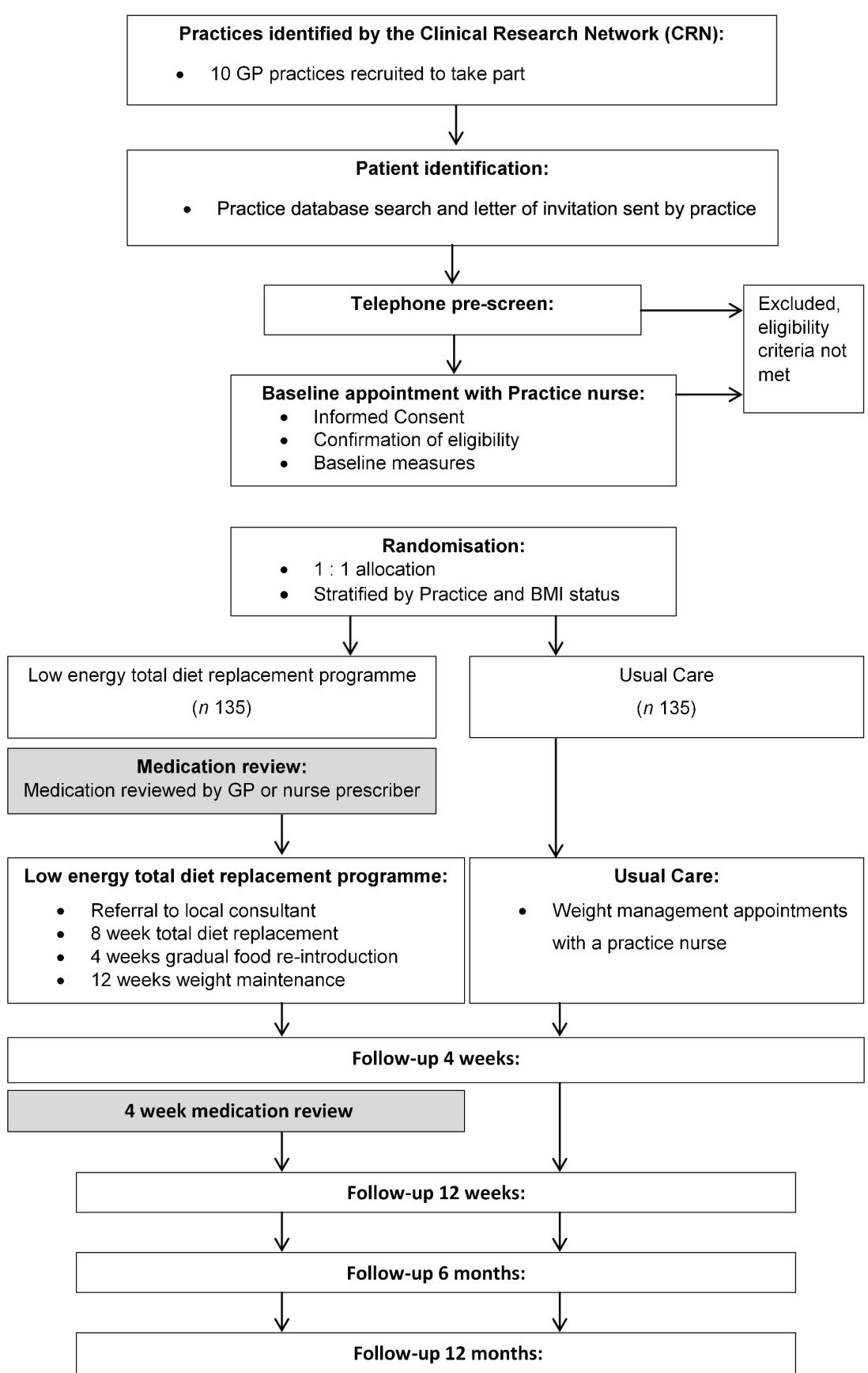

**Figure 1** Participant flow through the study. BMI, body mass index; GP, general practitioner.

| | VISIT | | | | |
|---|---|---|---|---|---|
| | Enrolment | Follow-up visits | | | |
| **TIMEPOINT** | *Baseline* | *4 weeks* | *12 weeks* | *6 months* | *12 months* |
| **ENROLMENT:** | | | | | |
| **Informed consent** | X | | | | |
| **Eligibility screen** | X | | | | |
| **Randomisation** | X | | | | |
| **INTERVENTIONS:** | | | | | |
| *Low energy total diet replacement programme* | | ⟶ | ⟶ | ⟶ | |
| *Usual Care* | | ⟶ | ⟶ | | |
| **ASSESSMENTS:** | | | | | |
| *Demographic* | X | | | | |
| *Medical History* | X | | | | |
| *Concomitant Medication* | X | X | X | X | X |
| *Height* | X | | | | |
| *Weight* | X | X | X | X | X |
| *Body composition* | X | X | X | X | X |
| *Waist circumference* | X | X | X | X | X |
| *Blood Pressure* | X | X | X | X | X |
| *Fasting blood sample* | X | | | | X |
| *Medication review* | X | X | | | |
| **QUALITATIVE INTERVIEWS** | | | X | | X |
| **QUESTIONNAIRES:** | | | | | |
| *Quality of life: (EQ-5D and OWLQOL)* | X | | X | X | X |
| *Programme adherence* | | X | X | X | X |
| *Programme feedback* | | X | X | X | |
| *OXFAB[18]* | | | X | X | |

**Figure 2** Schedule of measurements.[18]

of the practice nurse team who has been trained to offer a weight loss programme. The trial will take place only in practices where this is routine care. Participants allocated to the usual care group will not be prevented from attending other weight management groups if they choose to do so, but no NHS referrals to these schemes

will be offered during the trial. The practice nurse will give participants a copy of the booklet 'So you want to lose weight … for good'.[17] This 47-page booklet provides advice akin to a behavioural weight management programme. The aim is to produce a weight loss goal of 0.5–1 kg/week. It includes goal setting, advice on portion control and physical activity, other behavioural strategies, and monitoring and feedback on progress. Nurses will be asked to offer a programme for 12 weeks, at a frequency that is usually used in the practice (eg, weekly or biweekly).

### Physical activity

We recognise the importance of the role of aerobic and resistance exercise in facilitating weight loss and maintaining lean body mass to facilitate weight loss maintenance.

Participants randomised to the low-energy total diet replacement arm are given appropriate advice based on their previous exercise history, current ability and what is appropriate for their stage weight loss programme. Clinical guidelines in the UK emphasise the importance of advice to increase physical activity, and we would expect this to be incorporated into the control 'usual care' intervention.

### Outcomes

#### Primary outcome
► change in body weight from baseline to 12 months.

#### Secondary outcomes
► change in body weight from baseline at 3 and 6 months
► proportion of participants achieving 5% and 10% weight loss at 12 months
► change in fat mass between baseline and 12 months
► change in low-density lipoprotein (LDL) cholesterol concentrations between baseline and 12 months
► change in haemoglobin A1c (HbA1c) between baseline and 12 months
► change in systolic and diastolic blood pressure between baseline and 12 months.

#### Exploratory outcomes
► change in fat mass from baseline to 12 weeks and from baseline to 6 months
► change in waist circumference from baseline to 3, 6 and 12 months
► change in triglyceride and high-density lipoprotein (HDL) cholesterol concentrations between baseline and 12 months
► change in fasting glucose and insulin concentrations and change in Homeostatic Model Assesment (HOMA) of insulin resistance (HOMA-IR), insulin sensitivity (HOMA-%S) and beta cell function (HOMA-%B) between baseline and 12 months
► change in systolic and diastolic blood pressure between baseline and 3 months and between baseline and 6 months

► change in QRISK between baseline and 12 months
► change in the quality of Life measured using the EQ-5D scale between baseline and 12 months
► change in obesity-related quality of life measured with the Obesity-Specific Quality of Life (OWLQOL) between baseline, 3, 6 and 12 months
► proportion of people continuing their weight loss attempt and following the prescribed programme at 4, 8 and 12 weeks
► the number of weight control behaviours that participants are using assessed using the Oxford Food and Activity Behaviours (OxFAB) questionnaire[18] at 3 and 6 months
► qualitative interviews with a subsample of participants at 6 and 12 months
► adverse event (AE) reports up to 12 weeks, the end of the weight loss intervention or 6 months for AEs known or presumed to be related to gallstones.

### Measurements

Figure 2 provides a summary of the measurements collected.

### Sociodemographic characteristics

Participants will be asked to self-report age, sex and ethnicity.

### Medical history

Relevant medical history and all concomitant medication will be recorded and checked against the participants' medical record. Participants will also be asked to self-report items required to determine cardiovascular risk score using QRISK2.[19]

### Physical measurements

Height will be measured to the nearest 1 cm using stadiometers available in the practice. Weight will be recorded to the nearest 0.1 kg using an electronic scale (SC-240 MA, Tanita Japan), which will also record the proportion of body fat using bioelectrical impedance. Waist circumference will be measured in the horizontal plane at the upper border of the iliac crest at the end of expiration[20] using a fibreglass non-stretch tape measure fitted with a tensioning device (Gulick II Tape Measure, Fitness Mart USA). Seated blood pressure will be measured in triplicate with 1 min between each measure. All physical measures are performed by assessors trained according to the study manual of procedures.

### Fasting blood sample

A fasting venous blood sample will be collected (to be analysed for glucose, insulin, HbA1c, HDL and LDL cholesterol, triglycerides). When baseline/enrolment appointments are scheduled at times when it may be inappropriate to fast, participants will be asked to arrange for a fasting blood sample to be collected at an alternative appointment within 7 days of the enrolment visit and before the participant commences the allocated weight loss programme.

## Questionnaires

Participants will be provided with a questionnaire booklet which they will be asked to complete and return to the trial team in a postage paid envelope provided. The questionnaire booklet contains the following measures:

► *OWLQOL*: a weight-specific instrument intended to be used to assess obesity-specific symptoms and quality of life, general functional status and well-being, and person-specific preference measurement.[21]

► *Quality of life*: EQ-5D will be used as a standardised validated instrument used for measuring general health status.[22]

► *Programme adherence*: self-reported adherence to the allocated programme and methods participants are using to attempt to lose weight will be recorded by questionnaire.

► *Programme feedback*: will be assessed using several 5-point Likert scales, including whether there is an aim to continue with the programme.

► *OxFAB*: a questionnaire to assess personal strategies used by individuals for the purposes of weight loss.[18]

## Retention and withdrawal

We will seek to follow up all participants except those who expressly withdraw from the study. Participants who decide to withdraw from or discontinue the intervention allocated as part of the study will be asked to return for follow-up visits to collect outcome measures. To promote participant retention and complete follow-up, participants will be offered a £10 gift card for attending each of the 6-month and 12-month follow-up visits.

## Adverse events

AEs are of relevance in this trial because many practitioners feel these programmes are poorly tolerated and unsuitable for routine use in primary care. We will record AEs following Good Clinical Practice. All serious and non-serious AEs that occur during the first 12 weeks of the study or until the termination of the weight loss programme will be recorded in participants who initiate one of the weight loss interventions. We will also record all AEs that are presumed to be or known to be related to gallstones up to 6 months.

## Data management

Data will be recorded in a web-based data capture system (OpenClinica), which is hosted by the Primary Care Clinical Trials Unit of the University of Oxford. This system is customised and has an audit trail facility. Ranges and programmed validation checks are implemented in the system in order to aid reliable data entry.

## Statistical analysis

The primary and secondary outcomes will be assessed using an intention-to-treat analysis by an independent statistician. Each continuous outcome will be assumed to follow the normal distribution and be analysed by means of a linear mixed-effects model, adjusted for outcome at baseline. The model will include fixed-effects terms for randomised group, visit, interaction between randomised group and visit, and baseline BMI (for non-weight outcomes only), and random effects to account for repeated measures on the same participant at 3, 6 and 12 months. No adjustment will be made for baseline BMI in the analysis of the weight outcomes due to its strong collinearity with baseline weight. A random effect will also be included for individual practice. An unstructured variance–covariance matrix will be specified between repeated measurements on the same individual, and the random effects for patient and practice will be assumed to be independent. The adjusted treatment effect together with the 95% CI and p value will be reported. The analysis will be performed using PROC MIXED in SAS Version 9.4. The proportion of participants who lose 5% and 10% of their initial weight at 12 months, respectively, will be presented, and the adjusted difference between the two arms and 95% CI will be reported. The binary outcome will be analysed by means of a logistic mixed-effects model, adjusting for baseline BMI (fixed effect) and practice (random effect). The number needed to treat to achieve 5% or 10% weight loss, defined as the inverse of the absolute difference in proportions, will be reported if the differences between the treatment and control groups are statistically significant. A full statistical analysis plan will be prepared prior to any data analysis.

## Qualitative substudy

The purpose of this study is to examine participants' views of the programmes. In particular, we aim to examine the features that helped or hindered adherence to the programme and participants' views of the behavioural support provided in the respective programmes. We will therefore purposively sample participants based on their responses to the satisfaction questionnaire, reflecting positive, neutral and negative evaluations. Where possible, we will select participants to reflect both genders, socioeconomic status and ethnic group differences. We anticipate interviewing around 20 participants in the intervention group and 10 in the control group, but sampling will continue until saturation is reached, evidenced by no new themes occurring.

We will develop a semistructured topic guide for the interviews. The interviewer will encourage respondents to discuss their perceptions and experiences freely and in depth. The interview will set the context by asking about previous experience of weight management. Thereafter, we will ask for participants' views on which component parts of their treatment they felt were effective and which they felt were not effective; thoughts about ability to continue to manage their weight when treatment has ended; and their views on medication adjustments where these occurred. The acceptability of the weight management treatment programmes and any preference they initially had for the total diet replacement programme or the usual care programme will be explored.

Data from participants will be collected in a confidential, telephone interview, which will be audio-recorded.

All interviews will be transcribed. To examine saturation, analysis will proceed concurrently with interviewing.

## Trial Steering Committee

An independent Trial Steering Committee (TSC) will provide oversight of all matters relating to participant safety and data quality and value to the public. Due to the low risk nature of the DROPLET trial and that it is an open-label trial, the TSC also has the role of the Data Monitoring Committee, in addition to its role as the TSC. However, there are no early stopping rules, and all AEs are evaluated unblinded to allocation by the trial management group as well as the TSC.

The TSC includes an independent clinician, dietitian, statistician and two patient representatives. The TSC has reviewed the trial protocol, statistical analysis plan and the suitability of the proposed safety data to be collected. No interim analysis is planned for this trial due to the short recruitment period and low risk nature of the two dietary approaches.[11] The trial may be subject to inspection and audit by the University of Oxford, under their remit as sponsor, the trial coordinating centre as the Sponsor's delegate and other regulatory bodies.

## Ethics and dissemination

The study protocol (V.4.0; 5 October 2016) was reviewed and approved by the South Central Oxford B REC Committee (Ref: 157/SC/0337). Any protocol modifications will be sent for review by the research ethics committee and will be amended at the trial registry.

It is planned that results will be disseminated to academic and health professional audiences via presentations at conferences and publication in peer-reviewed journals. Participants will be sent a summary of the trial findings at the time when the main article is published. If the trial shows this intervention is effective, the results will be communicated to policymakers and commissioners of weight management services through briefing papers summarising the main findings. We will also provide the results to all participants coincident with publication and disseminate the results to the public through a press release, regardless of what the results show.

**Acknowledgements** The low-energy total diet replacement programme including the formula meal replacement products will be provided by Cambridge Weight Plan, Northants, UK.

**Contributors** SAJ and PA designed the study and secured the funding. NMA and ST helped to develop the protocol. NMA is the trial manger and AN is the trial statistician.

**Funding** This research is funded by research grants from Cambridge Weight Plan Ltdand NIHR Collaboration forLeadership in Applied Health Research and Care (CLAHRC) Oxford at Oxford HealthNHS Foundation to the University of Oxford. The sponsor of the trial isthe University of Oxford. The protocol was initiated and designed by the investigators who have nopersonal financial relationships with the Cambridge Weight Plan Ltd. Although Cambridge Weight Plan were consultedand commented on the protocol, the final decisions lay with the investigators.There are no restrictions on publication of results arising from this study andthe contract between the funder and the University ensures that the fundingbody will have no input into the decisions regarding publication.

**Competing interests** SAJ and PA have led publicly funded trials in which the weight management intervention was provided free of charge by other commercial companies. They receive no personal financial benefits from these trials. NMA,

ST and AN have no competing interests. Cambridge Weight Plan, as the funder of this trial, is also the manufacturer of the nutritional products used in the trial and provided the products used in the trial free of charge to the participants.

**Ethics approval** NHS Research Ethics Committee (South Central Oxford B Committee).

**Provenance and peer review** Not commissioned; externally peer reviewed.

**Data sharing statement** For access to the data set, a formal request should be sent to theDROPLET study group. The request will only be considered when the principalresults of the study have been published.

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
