## [Reviewer comments · BMJ Open]

ARTICLE DETAILS

TITLE (PROVISIONAL)	Doctor referral of overweight people to a low-energy treatment (DROPLET) in primary care using total diet replacement products: a protocol for a randomised controlled trial
AUTHORS	Jebb, Susan; Astbury, Nerys; Tearne, Sarah; Nickless, Alecia; Aveyard, Paul

VERSION 1 - REVIEW

REVIEWER	Mike Lean University of Glasgow Scotland PI on the DiRECT trial (not exactly competing, but interested)
REVIEW RETURNED	10-May-2017

GENERAL COMMENTS	This is a welcome study protocol, adding to the limited research in an area of great medical need. I have a few comments which should be considered by the authors. 1. Abstract (and elsewhere) I feel that mean weight change is not really appropriate for this sort of realistic study. Would not the % with categorical changes be more meaningful, and more amenable to ITT analysis? We do not need more evidence that TDR is superior to conventional weight-management advice in primary care, but ITT analysis and economic evaluations are vital. 2. Strengths and Limitations. (i) This study is not the first RCT of TDR in routine primary care. Unless the authors are quibbling over RCT vs CRT, DiRECT might hold that position, but will not be bragging about it. (ii) The distinction in the second bullet point is between delivery by a commercial provider using non-NHS staff in DROPLET, as distinct from delivery by existing trained NHS staff within routine primary care. (iii) Final bullet point seems unnecessary and confusing. This is not the only possible source of bias, and it might be argued that NHS staff are better placed to guide patients than non-NHS lay counsellors. 3. Line 88-90. This is making several assumptions. These patients are high attenders at GP surgeries, and the service need to provide effective weight management may in fact require increased or excessive overall input. Interesting. 4. Line 89-90. It is brave to claim that referral to a commercial provider is the most effective option. It is one effective option, but
---

this paper references the Counterweight-Plus feasibility study, which used existing NHS staff, and might be regarded as more effective.

5. Line 97. The current composition of TDRs is required to include 100% or the DRVs for essential nutrients. It is important to reflect that those DRVs are designed for people who are healthy and weight-stable. During weight loss on TDR, requirements change. Protein turnover is reduced, so the need for protein is reduced. The need for other micronutrients are reduced, probably including essential fatty acids. For some micronutrients, there is benefit from including more than the DRV, for example magnesium, because so many obese patients (especially with type 2 diabetes) are relatively deficient in magnesium, and because without it constipation can be a real problem. The lack of dietary fibre in TDR feeds is worth mentioning too: a supplement is usually required.

5, Line 100-. The comparator diets in that meta-analysis were not representative of usual weight management in routine NHS primary care: many were very intensive (weight loss up to 11kg at 12 months). It would be more appropriate to use data from routine NHS care for the power analysis. The data from TDRs in research settings in that meta-analysis may also be inappropriate unless the results from the commercial delivery of Cambridge weight Plan are similar. This point might be worth discussing more, but the final figure of 4kg difference is probably roughly right!

6. Line 114. I think EFSA defines TDR for weight loss as under 1200 kcal/day.

7. Line 126-130. This is not correct. The Counterweight-Plus intervention in DiRECT is delivered by existing non-specialist NHS staff in routine primary care (practice nurses or dietitians if available), with very brief training. The evidence gap is whether referral to a commercial non-health professional lay service is effective, and cost-effective.

8. Line 216. Please specify what is meant by 'member of the clinical care team'. Is that the primary care team, for which the GP takes responsibility? Is there any advisory or other input from the researchers?

9. Line 251. Please define the amount and validation of the training of the Cambridge Weight Plan Counsellors (They are referred to as 'Consultants' elsewhere. That might confuse clients.)

10. Line 251. What is the disclosure status of the Counsellors? Is that an issue, if NHS patients might be considered vulnerable?

11. Line 251, Attend 'regular appointments for 24 weeks'. Please specify how many appointments in the second 24 weeks. Presumably it is not possible to specify the number of appointments will be offered to control patients at this stage.

12. Line 275-277. If this study truly aims to compare referral to Cambridge Weight Plan with usual weight management in primary care, then both groups must be permitted to engage in other weight-control activities (they will do that anyway!). And if referral to another commercial service such as Weight Watchers is available, then that should be permitted. Some patients find that helpful while using TDRs.

	13. Line 287 and 290. I would argue that these outcomes should be reversed. 14. 291 and 322. Bioimpedance offers high, probably spurious, precision, and as far as I am aware it has never been shown to be superior to prediction of fat mass by published anthropometric equations. The estimation of lean body mass from bioimpedance is even less accurate, and potentially misleading given that muscle mass is what actually matters. Given that the anthropometric data (height, weight, waist, hips) are being, or can easily be, collected, it would make sense to incorporate in the Outcomes estimations of adipose fat mass and of whole body muscle mass from published equations. 14. Line 409. An oversight is what leads to substantial amendments of protocols! The section might be better understood as 'Trial Management'.
--	---

REVIEWER	James Hill University of Colorado Denver, CO USA I have an equity interest in two companies (Retrofit, Shakabuku) that provide weight management services on a fee-for-service basis.
REVIEW RETURNED	15-May-2017

GENERAL COMMENTS	This manuscript reports a protocol that was developed for conducting a randomized, controlled trial comparing two different weight management programs. The trial will compare a commercial program using very low energy diets (VLEDs) to a behavioral program delivered in the form of a booklet by a member of the practice groups. The rationale for using VLEDs for weight loss is well presented. The protocol is very reasonable and makes total sense for a randomized trial. My only comment on the protocol is that there is not any specific information about physical activity. While physical activity may not contribute much to weight loss, it does seem to be important for weight loss maintenance. What is the physical activity protocol that will be used and what are the goals for physical activity? The protocol may already be set, but I would submit the following comments for consideration:  1. Is this trial really going to add much to what is already in the literature? According to the review paper cited, there have been 6 trials of 24 months duration with VLEDs providing an average weight loss of 4.2 kg – 1.4 kg more than behavior therapy alone. Will 100 more subjects (assuming drop outs) really add much? Previous trials are criticized as being relatively small (30-100 subjects) but this one will not be much larger (n=135 per group will be recruited). 2. The comparator in this study is a behavioral program delivered by booklet. I think we know that this program will not provide very much weight loss. Wouldn't it be better to compare it to another commercial weight loss program that does not involve VLEDs – such as Weight Watchers or Slimming World? 3. If commercial programs could be used for both arms of the trial, subjects could be followed for longer periods of time – maybe for 5 years. We have very few prospective trials of this duration.
---

VERSION 1 – AUTHOR RESPONSE

Reviewer 1

This is a welcome study protocol, adding to the limited research in an area of great medical need. Thank you.

I have a few comments which should be considered by the authors.

1. Abstract (and elsewhere) I feel that mean weight change is not really appropriate for this sort of realistic study. Would not the % with categorical changes be more meaningful, and more amenable to ITT analysis? We do not need more evidence that TDR is superior to conventional weight-management advice in primary care, but ITT analysis and economic evaluations are vital.

In our systematic review of randomized controlled trials of VLED interventions all the studies were conducted in specialist clinics or research settings. There were none in primary care and none in which the intervention was provided by lay staff. We firmly believe that the importance of this trial is to test whether the results seen in previous studies can be replicated in a generalist setting with the intervention delivered in a manner which could be rolled out at scale. We know that there has been a significant dilution of the effectiveness of many other weight loss interventions when translated from a specialist setting to routine care. We have set out plans for an ITT analysis and will report data as in our previous trials of weight loss where the primary outcome was also mean weight change. We consider that mean weight change is the most appropriate primary outcome, though we also include categorical changes (5 and 10% losses) as secondary outcomes. In most weight-loss interventions the two approaches are strongly related since weight losses are normally distributed making it possible to estimate the proportion losing 5 or 10% from the mean and SD of weight change. Moreover, mean weight change is an outcome common to almost all weight-loss trials and hence our data can be readily compared with other interventions. Finally, we have pre-registered the trial with mean weight change as the primary outcome and, while this could be changed, we fear it will cast unnecessary doubt on the integrity of the analysis to change at this stage given that there is no specific case against mean weight loss as the primary outcome.

Aveyard P et al (2016) Screening and brief intervention for obesity in primary care: a parallel, two-arm, randomised trial. *The Lancet* 388(10059):2492-25000

Ahern et al (2017) Extended and standard duration weight-loss programme referrals for adults in primary care (WRAP): a randomised controlled trial. *The Lancet* 389(10085):2214-2225

2. Strengths and Limitations.

(i) This study is not the first RCT of TDR in routine primary care. Unless the authors are quibbling over RCT vs CRT, DiRECT might hold that position, but will not be bragging about it.

The DiRECT trial is recruiting patients with type 2 diabetes and has remission of diabetes and weight loss > 15% as co-primary outcomes. The DiRECT investigators make the case that the potential for remission of their diabetes provides a powerful motivator which promotes adherence to the programme. In the DROPLET trial we recruit patients simply on the basis of having a BMI >30 and offer the programme as a first line treatment for routine weight management for patients with or without obesity related co-morbidities. Nonetheless we have softened the wording in the paper which now reads:

“This study is the largest randomised controlled trial to date of a low-energy total diet replacement programme for weight management in routine primary care”

(ii) The distinction in the second bullet point is between delivery by a commercial provider using non-NHS staff in DROPLET, as distinct from delivery by existing trained NHS staff within routine primary care.

We have amended as suggested. The bullet point now reads:

“This intervention is based on a model of care where GPs refer patients to a programme delivered in

the community by a commercial provider using non-NHS staff, which, if successful, could be readily adopted into practice without the need for specialist training for the primary care workforce.”

(iii) Final bullet point seems unnecessary and confusing. This is not the only possible source of bias, and it might be argued that NHS staff are better placed to guide patients than non-NHS lay counsellors.

We have deleted this bullet point.

3. Line 88-90. This is making several assumptions. These patients are high attenders at GP surgeries, and the service need to provide effective weight management may in fact require increased or excessive overall input. Interesting.

We state that;

“However, although a number of interventions have been shown to be effective in intensive research studies, this success has not always been replicated in routine settings.”

In this trial we recruit by sending a letter to all patients with a recorded BMI >30. We have found that many of these people are not regular attendees in primary care. This is an important distinction from studies like DiRECT, which specifically recruit patients with type 2 diabetes. We accept that people who are obese ought to be receiving more intensive care, but in the current NHS climate this seems hard to achieve. So here we seek to observe whether health benefits can be achieved by referring patients from primary care to a community provider.

4. Line 89-90. It is brave to claim that referral to a commercial provider is the most effective option. It is one effective option, but this paper references the Counterweight-Plus feasibility study, which used existing NHS staff, and might be regarded as more effective.

We cite evidence from two systematic reviews of trials which observed that interventions delivered by generalist primary care staff were ineffective and data from several trials and systematic reviews, including thousands of participants, shows significantly greater weight loss following referral to a commercial provider running community based open weight loss groups. This approach is the mainstay of Tier 2 weight management services in the UK. The Counterweight-Plus feasibility study was not a trial and involved only 91 patients and did not include a comparator group. While promising, we consider it premature to claim it is more effective. However we have amended the wording to remove the suggestion that weight loss groups are the “most effective” option. This sentence now reads:

“GP referral to a commercial provider offering group-based support is an effective option for weight management in primary care, and our meta-analysis showed a mean reduction in weight of 2.3 kg over no intervention at one year”.

5. Line 97. The current composition of TDRs is required to include 100% of the DRVs for essential nutrients. It is important to reflect that those DRVs are designed for people who are healthy and weight-stable. During weight loss on TDR, requirements change. Protein turnover is reduced, so the need for protein is reduced. the need for other micronutrients are reduced, probably including essential fatty acids. For some micronutrients, there is benefit from including more than the DRV, for example magnesium, because so many obese patients (especially with type 2 diabetes) are relatively deficient in magnesium, and because without it constipation can be a real problem. The lack of dietary fibre in TDR feeds is worth mentioning too: a supplement is usually required.

Thank you. We have amended to read:

“When used as directed, these formula products meet 100% of the dietary reference values for vitamins, minerals and trace elements for healthy, weight-stable people and are enriched with high biological-value protein. Although most contain some dietary fibre, a fibre supplement may also be recommended.”

5. Line 100-. The comparator diets in that meta-analysis were not representative of usual weight management in routine NHS primary care: many were very intensive (weight loss up to 11kg at 12 months). It would be more appropriate to use data from routine NHS care for the power analysis. The data from TDRs in research settings in that meta-analysis may also be inappropriate unless the results from the commercial delivery of Cambridge weight Plan are similar. This point might be worth discussing more, but the final figure of 4kg difference is probably roughly right!

Since there is no previous trial of this kind in routine primary care we have had to use some judgement in the sample size calculations. We do not consider it appropriate to use data from usual delivery of the Cambridge Weight Plan since this refers exclusively to people who self-select and self-fund attendance at this form of weight management and published data refers only to those who 'complete' the programme. We have set out the basis of our calculations and we are pleased to note that the reviewer concurs that this is "probably roughly right". Coincidentally, it is very comparable to the sample size in the DiRECT trial and larger than any previous trial of TDR.

6. Line 114. I think EFSA defines TDR for weight loss as under 1200 kcal/day. We have amended accordingly.

7. Line 126-130. This is not correct. The Counterweight-Pus intervention in DiRECT is delivered by existing non-specialist NHS staff in routine primary care (practice nurses or dietitians if available), with very brief training. The evidence gap is whether referral to a commercial non-health professional lay service is effective, and cost-effective.

We consider that dietitians do have specialist diet training. Nonetheless the reviewer is correct in identifying that the key difference is probably the use of NHS vs non-NHS staff and we have amended the text to this effect.

"It will be delivered by NHS staff whereas the present study will test the effectiveness of referral outside the NHS to a commercial provider."

8. Line 216. Please specify what is meant by 'member of the clinical care team'. Is that the primary care team, for which the GP takes responsibility? Is there any advisory or other input from the researchers?

Guidance on medication adjustments is provided by the study team for use of prescribers. We have amended to indicate this is usually a GP or a trained nurse prescriber. This guidance is provided in the supplementary material as already detailed. Thus:

"Participants allocated to the low-energy total diet replacement programme and taking medications for type 2 diabetes, hypertension or high cholesterol will have their medications reviewed by a prescribing member of the clinical care team, usually the GP or trained nurse prescriber. During this medication review the clinician will decide what changes to medications are required at the time the participant commences the low energy total diet replacement programme, with guidance provided by the study team (Supplementary Figure 1)."

9. Line 251. Please define the amount and validation of the training of the Cambridge Weight Plan Counsellors (They are referred to as 'Consultants' elsewhere. That might confuse clients.)

We have ensured that we are consistent in the use of the term counsellor and have added additional detail as requested:

"All counsellors attend a 1-day in-person training course covering screening for suitability, nutrition, behavioural approaches, and medical monitoring. They must pass an accreditation examination before they are allowed to deliver the programme in the community. Thereafter, they have a yearly training updates, a nominated sponsor and access to an online chat forum for sharing queries. Cambridge Weight Plan has a healthcare professional available for the counsellors to consult for advice on specific medical and nutritional queries. Counsellors delivering the intervention for the purposes of this trial received short trial specific training before being allocated study participants. "

10. Line 251. What is the disclosure status of the Counsellors? Is that an issue, if NHS patients might be considered vulnerable?

Currently counsellors delivering the total diet replacement intervention are not required to undergo a DBS check.

11. Line 251, Attend 'regular appointments for 24 weeks'. Please specify how many appointments in the second 24 weeks. Presumably it is not possible to specify the number of appointments will be offered to control patients at this stage.

We have added a sentence with this additional information.

“after 12 weeks participants attend monthly appointments at 16, 20 and 24 weeks”

12. Line 275-277. If this study truly aims to compare referral to Cambridge Weight Plan with usual weight management in primary care, then both groups must be permitted to engage in other weight-control activities (they will do that anyway!). And if referral to another commercial service such as Weight Watchers is available, then that should be permitted. Some patients find that helpful while using TDRs.

We state that “Participants allocated to the usual care group will not be prevented from attending other weight management groups if they choose to do so, but no NHS referrals to these schemes will be offered during the trial.” In the outcome measures we indicate that we will collect information on weight control practices.

Since we are trying to establish the effectiveness of a total diet replacement programme relative to interventions delivered by practice nurses we consider it reasonable not to offer alternative interventions to either group for the duration of the trial. But we acknowledge and will not inhibit personal decisions to use an alternative programme and we will record any such actions. In our experience, very few people choose to pay for attendance at an alternative programme during the course of the trial.

13. Line 287 and 290. I would argue that these outcomes should be reversed.

Please see our response to point 1. Abstract (and elsewhere), above.

14. 291 and 322. Bioimpedance offers high, probably spurious, precision, and as far as I am aware it has never been shown to be superior to prediction of fat mass by published anthropometric equations. The estimation of lean body mass from bioimpedance is even less accurate, and potentially misleading given that muscle mass is what actually matters. Given that the anthropometric data (height, weight, waist, hips) are being, or can easily be, collected, it would make sense to incorporate in the Outcomes estimations of adipose, fat mass and of whole body muscle mass from published equations.

Estimates of fat mass are secondary outcomes. Bioimpedance relies heavily on age, gender, height and weight and we concur with the reviewer that neither bioimpedance nor anthropometric measures reliably measure muscle mass. We will use scales for weight measures which have the capacity to measure bioimpedance but since we have no specific hypothesis for any differential effect of the two treatments on the composition of weight lost we do not feel it is appropriate to add further secondary outcomes.

14. Line 409. An oversight is what leads to substantial amendments of protocols! The section might be better understood as 'Trial Management'.

We accept that 'oversight' is perhaps not the best word, but neither is management. The trial management refers to the day to day conduct of the trial. In this section of the protocol we describe the additional arrangements which involve external scrutiny. We have renamed this “Trial Steering Committee”.

Reviewer: 2

This manuscript reports a protocol that was developed for conducting a randomized, controlled trial comparing two different weight management programs. The trial will compare a commercial program using very low energy diets (VLEDs) to a behavioral program delivered in the form of a booklet by a member of the practice groups. The rationale for using VLEDs for weight loss is well presented. The protocol is very reasonable and makes total sense for a randomized trial. Thank you.

My only comment on the protocol is that there is not any specific information about physical activity. While physical activity may not contribute much to weight loss, it does seem to be important for weight loss maintenance. What is the physical activity protocol that will be used and what are the goals for physical activity?

The physical activity guidance will be specific to the treatment arm.

We have now included detail of the advice for physical activity provided in the two interventions:

“We recognise the importance of the role of aerobic and resistance exercise in facilitating weight loss and maintaining lean body mass to facilitate weight loss maintenance.

Participants randomised to the low-energy total diet replacement arm are given appropriate advice based on their previous exercise history, current ability and what is appropriate for their stage weight loss programme. Clinical guidelines in the UK emphasise the importance of advice to increase physical activity and we would expect this to be incorporated into the control ‘usual care’ intervention.”

The protocol may already be set, but I would submit the following comments for consideration:

1. Is this trial really going to add much to what is already in the literature? According to the review paper cited, there have been 6 trials of 24 months duration with VLEDs providing an average weight loss of 4.2 kg – 1.4. kg more than behavior therapy alone. Will 100 more subjects (assuming drop outs) really add much? Previous trials are criticized as being relatively small (30-100 subjects) but this one will not be much larger (n=135 per group will be recruited).

Although VLEDs are increasingly used in intensive research settings and specialist clinics, the SR of their effect found that there were only 12 trials with >12 month follow-up. These were largely efficacy studies conducted by highly trained staff in specialist research settings. Although this trial is larger, the key distinction from the previous studies is that we will use TDR in routine settings to evaluate the effectiveness of the approach for use by non-specialists.

2. The comparator in this study is a behavioral program delivered by booklet. I think we know that this program will not provide very much weight loss. Wouldn't it be better to compare it to another commercial weight loss program that does not involve VLEDs – such as Weight Watchers or Slimming World?

The reviewer has misunderstood the nature of the comparator treatment which will be a nurse-led face-to-face weight management programme, which represents the “usual care”. The content will be based on a booklet which we have shown to be associated with weight loss in our other trials eg. Ahern et al 2017 where it was associated with 3.26kg weight loss at 1 year. This booklet is commonly used as a guide for weight management delivered in routine care in the UK, though here, the general content of the booklet will be delivered by means of face-to-face consultations over 12 weeks.

We say:

“The comparator intervention will consist of the usual weight management programme provided by a member of the practice nurse team who has been trained to offer a weight loss programme. The trial will take place only in practices where this is routine care..... Nurses will be asked to offer a programme for 12 weeks, at a frequency that is usually used in the practice (e.g. weekly or bi-weekly).”

We already know that WeightWatchers and Slimming World are effective for weight loss. Here we aim

to see if the Cambridge Weight Plan is effective. If so it could be offered as an alternative option in primary care. If effective, subsequent studies could seek to compare the two treatments or to identify which participants may be best suited to which programme, but first we need to show this programme is effective in the context of routine weight management, with relatively unselected participants and delivered by a commercial provider and not NHS staff or university researchers.

3. If commercial programs could be used for both arms of the trial, subjects could be followed for longer periods of time – maybe for 5 years. We have very few prospective trials of this duration. The duration of follow up is independent of the programme. Commercial programmes usually only follow up people who continue to attend. Ideally, by recruiting patients through primary care it would be possible to use data linkage to follow them up through health records. However, we know that routine recording of weight is poor, so an extended follow up would require us to contact patients to attend an appointment at the surgery to be weighed. This would be possible if resources allowed but this is beyond the scope of the current trial, which first seeks to establish whether weight differs at 1 year. Weight changes will be measured over 1 year, however we acknowledge that longer term data would be helpful to better estimate the longer health impact and cost effectiveness of the intervention over longer term.

We have now noted this as a limitation of the trial:

“Longer term follow-up data would be helpful to better estimate the longer health impact and cost effectiveness of the intervention”

We look forward to hearing your response in due course